# Remotely Sensed Phenotypic Traits for Heritability Estimates and Grain Yield Prediction of Barley Using Multispectral Imaging from UAVs

**DOI:** 10.3390/s23115008

**Published:** 2023-05-23

**Authors:** Dessislava Ganeva, Eugenia Roumenina, Petar Dimitrov, Alexander Gikov, Georgi Jelev, Boryana Dyulgenova, Darina Valcheva, Violeta Bozhanova

**Affiliations:** 1Space Research and Technology Institute, Bulgarian Academy of Sciences, 1113 Sofia, Bulgaria; roumenina@space.bas.bg (E.R.); petar.dimitrov@space.bas.bg (P.D.); gikov@space.bas.bg (A.G.); gjelev@space.bas.bg (G.J.); 2Institute of Agriculture, Agriculture Academy, 8400 Karnobat, Bulgaria; bdyulgerova@abv.bg (B.D.); darinadv@abv.bg (D.V.); 3Field Crops Institute, Agricultural Academy, 6200 Chirpan, Bulgaria; bozhanova@agriacad.bg

**Keywords:** barley, grain yield prediction, heritability, multispectral images, phenotypic traits, UAV

## Abstract

This study tested the potential of parametric and nonparametric regression modeling utilizing multispectral data from two different unoccupied aerial vehicles (UAVs) as a tool for the prediction of and indirect selection of grain yield (GY) in barley breeding experiments. The coefficient of determination (*R*^2^) of the nonparametric models for GY prediction ranged between 0.33 and 0.61 depending on the UAV and flight date, where the highest value was achieved with the DJI Phantom 4 Multispectral (P4M) image from 26 May (milk ripening). The parametric models performed worse than the nonparametric ones for GY prediction. Independent of the retrieval method and UAV, GY retrieval was more accurate in milk ripening than dough ripening. The leaf area index (LAI), fraction of absorbed photosynthetically active radiation (fAPAR), fraction vegetation cover (fCover), and leaf chlorophyll content (LCC) were modeled at milk ripening using nonparametric models with the P4M images. A significant effect of the genotype was found for the estimated biophysical variables, which was referred to as remotely sensed phenotypic traits (RSPTs). Measured GY heritability was lower, with a few exceptions, compared to the RSPTs, indicating that GY was more environmentally influenced than the RSPTs. The moderate to strong genetic correlation of the RSPTs to GY in the present study indicated their potential utility as an indirect selection approach to identify high-yield genotypes of winter barley.

## 1. Introduction

The development of new crop varieties, displaying greater yield potential and stress tolerance, is critical for dealing with the expected increase in food demands of the global population [1,2]. Phenotyping is a key methodological approach in the breeding process for the selection of improved varieties [3]. Phenotyping is the process of characterizing the phenotype where phenotype, as defined by Pieruschka and Poorter (2012) [4], is the “combination of all the morphological, physiological, anatomical, chemical, developmental and behavioural characteristics that, when put together, represent the individual organism” (p. 1). In breeding practice, phenotyping includes measurement or visual evaluation by experts of important performance- or quality-related plant traits such as height, biomass or grain yield (GY), phenology, etc. Phenotyping is a time and labor-intensive process when traditional and manual methods are used and it can be a significant challenge in the current breeding programs, where it is not uncommon to have over a thousand breeding lines with replications across multiple environments [5]. Low efficiency in collecting data to characterize the phenotype over such large populations is considered a major barrier to future breeding advances [1,5]. Therefore, new approaches for high-throughput phenotyping (HTP) have been proposed in recent years, which allow for the rapid, non-destructive, and accurate measurement of traits using different proximal/remote sensors [6]. In addition, due to its ease and the reduced time for data collection, HTP allows for multiple observations to be made during the growing season, thus tracking plant responses over time [5]. HTP has been implemented in both laboratory environments (e.g., greenhouses or growth chambers [7]) and the field.

Different platforms have been demonstrated useful for field-based HTP, including tractors [5] and unmanned aerial vehicles (UAVs) [8,9]. The main advantage of UAVs compared with tractor-based platforms is that all plots within a trial are measured almost simultaneously [1]. UAVs can carry various sensors, such as LIDAR (light detection and ranging) [10], thermals [11], and hyperspectral cameras [12], but low-cost red-green-blue (RGB) and multispectral cameras are common alternatives (for a recent review of sensors and applications of UAVs in HTP, see [13,14]). Depending on the type of images, different information can be extracted. For example, high-resolution images from RGB cameras can be used to reconstruct the 3D structure of the plant canopy and thus allow directly measuring morphology traits, such as height [9]. Multispectral and hyperspectral cameras collect spectral reflectance data at different wavelengths through the photosynthetically active radiation (PAR, 400–700 nm), near-infrared radiation (NIR, 700–1300 nm), and shortwave infrared (up to 3000 nm) regions of the electromagnetic spectrum [15]. Plant leaf reflectance patterns are characteristics of each of these regions. Mathematical combinations of reflectance values from different wavelength bands, i.e., vegetation indices (Vis) [16], can provide specific information for plant characteristics and states, such as water content [17] and chlorophyll [18]. The spectral reflectance and Vis can be regarded as “spectral traits”, which provide an integrated measurement of canopy structure and photosynthetic activity [8]. However, they are usually used to predict other agronomic traits to which they are well-correlated, e.g., yield, canopy cover, biomass, and leaf area index. Based on this correlation, prediction models are developed for the traits of interest.

While Vis are useful indicators of crop growth and condition, they are more difficult to interpret than biophysical variables such as the leaf area index (LAI), fraction of absorbed photosynthetically active radiation (fAPAR), fraction vegetation cover (fCover), leaf chlorophyll content (LCC), etc., which are traditionally used in crop studies and have well-understood relations with the physiological processes in plants, agronomic practices, and the environment. Biophysical variables provide valuable information about different aspects of crop state that can indicate productivity. For example, fAPAR has been used for drought monitoring [19] and LCC can be directly related to plant stresses, senescence, and nutritional state [20]. Previous studies have demonstrated that biophysical variables such as the LAI, fAPAR, and fCover are related to grain yield and can be used for its prediction [21,22,23,24,25].

Yield is a complex, quantitative trait that is under polygenic control [4]. It is known to be subject to low heritability and high genotype-by-environment interaction [9,26]. Therefore, replicated yield trials in the field are essential when selecting the yield as they allow for assessing plants in real-life conditions. The availability of an easy, rapid, and inexpensive selection tool to screen large numbers of genotypes before initiating expensive yield trials could reduce cost and time [27,28]. For this aim, indirect selection approaches are sought, which are based on secondary traits, namely traits that exhibit strong correlations with GY. Both aerial [8] and field-based [27] spectral measurements have been proved useful for indirect selection for GY. In addition, Cabrera-Bosquet et al. (2012) [3] point out that predictions of yield at early stages (e.g., before flowering) may speed up the design of test crosses at anthesis, saving time and cost. Moreover, Araus et al. (2008) [2] suggest that complementing traditional (i.e., direct selection for yield) with analytical (i.e., indirect selection for secondary traits) selection methodologies may be required to further improve grain yields. The authors mentioned NDVI and digital images as inexpensive methods to derive information for such traits, particularly those related to biomass.

Several studies in recent years have explored the association between spectral data/VIs and GY in barley (*Hordeum vulgare* L.) in the context of HTP with tractors [29,30,31] and UAVs [9,32,33]. The main questions in these studies concern the relative performance of UAVs versus ground sensors (such as GreenSeeker, Trimble, Westminster, CO, USA), the selection of an appropriate VI showing the highest correlation with GY, and the effect of phenology on this relationship.

The utilization of certain variables derived from remotely sensed data enables the forecasting of crop yields and retrieval of biophysical variables. This is possible because of the underlying mechanisms of how light interacts with leaves and canopy characteristics [34,35,36,37]. Consequently, diverse methods for yield estimation and biophysical variable retrieval are delineated in the literature [38,39]. The most adopted methods are the statistical regression algorithms that are divided into parametric and nonparametric algorithms. While parametric models are easy to calibrate, the nonparametric models are more flexible.

Broad-sense heritability (H^2^) is an important genetic parameter in crop breeding [40], as it allows plant breeders to estimate the degree to which genetic factors are responsible for variation in a particular trait. This information can then be used to make decisions about which breeding strategies to use to improve the trait [41].

In crop breeding, broad-sense heritability is typically used to estimate the genetic potential of a population for a given trait [42]. For example, if the H^2^ of a crop trait is high, it suggests that the trait is strongly influenced by genetic factors, and, therefore, breeding efforts should focus on identifying and selecting the individuals with the best genetic potential for the trait. Conversely, if the H^2^ of a trait is low, it suggests that the trait is strongly influenced by environmental factors, and breeding efforts should focus on identifying and selecting individuals with the best environmental adaptation. In remote sensing studies, remotely sensed traits are used to analyze the broad-sense heritability [43,44,45]. Additionally, heritability is used as feature selection technics for grain yield prediction [46,47].

The aim of the study is (1) to demonstrate the potential of using parametric and nonparametric regression models with multispectral data from UAVs as an indirect selection tool for grain yield in barley breeding experiments and (2) to estimate the proportion of phenotypic variability that is due to genetic factors using modeled biophysical variables and ground-measured NDVI.

## 2. Materials and Methods

### 2.1. Site (Field) and Experimental Design of the Study

The study was conducted during the 2020/2021 growing year in the experimental fields of the Institute of Agriculture, Karnobat, Southeastern Bulgaria (Figure 1). The soil of the experimental fields was slightly acidic (pH is 6.2) Pellic Vertisol.

Despite the total precipitation during the vegetation period (October 2020 to June 2021) being 197.4 mm higher than the long-term average precipitation for the location (424.6 mm), it was unevenly distributed throughout the growing season, as shown in Table 1. The amount of precipitation for January was the highest, 142.8 mm, which was 291% more compared to the long-term values for the location. The precipitation in April was 86 mm, which was nearly double the long-term sum for that month, followed by a significantly drier period in May. Except for March, April, and June, the monthly temperatures were higher than the average long-term temperatures for all other months of the growing season. Overall, the weather conditions enabled the distinguishing of productive abilities of tested barley genotypes.

Fifty-five genotypes of winter barley varieties and breeding lines were grown in rainfed field conditions. The genotypes were sown in two competitive variety trials (CVTs): CVT 1 and CVT 2. In CVT 1, 4 Bulgarian varieties (Emon, Obzor, Kaskadyor 3, and Dariya) and 21 advanced breeding lines of 2-rowed barley developed at the Institute of Agriculture, Karnobat were grown. CVT 2 included 30 genotypes: 1 Bulgarian 2-rowed variety (Emon), 1 Serbian 2-rowed variety (Sladoran), 4 Turkish 2-rowed varieties (Hasat, Harman, Bolayir, and Burgaz), 15 advanced 2-rowed breeding lines (KT 337, A 9/14, 167Д-2/05, 176Д-1/05, 419Д-2/08, 419Д-5/08, 530Д-2/09, 671Д-3/10, 718Д-4/10, 639Д-3/10, 003Д-3/13, 939Д-4/13, 194Д-1/15, 218Д-1/15, and WS270Д-1/15), 1 Italian 6-rowed variety (Futura), and 8 advanced 6-rowed breeding lines (KT 2207, KT 2213, KT 3040, KT 3041, KT 1706, KT 2199, SUE I, SUE II).

The experiments were organized in a complete block design with 4 replications on plots of 10 m^2^ with a sowing rate of 450 germinated seeds per m^2^. All genotypes included in CVT 1 and CVT 2 were sown on 25 October 2020. The standard technology for growing winter barley breeding materials at the Institute of Agriculture, Karnobat was applied. The predecessor was a pea–sunflower mix. One-time nitrogen (N) fertilization with a fertilizer rate of 30 kg/ha of active substance nitrogen in February 2021 was applied. The experiment was treated against weeds with a herbicide combination of Biathlon and Scorpio. No pesticides were used to control diseases or pests, as no pathogens and pests were observed at densities above the economic threshold values during the winter barley growing season.

### 2.2. Data Acquisition

#### 2.2.1. Measured Grain Yield (MGY)

All plants were harvested from each plot at maturity, threshed, and weighed to calculate the grain yield (kg/ha), Table 2.

#### 2.2.2. Field Measurements of Crop Biophysical Variables

A field campaign was conducted on 25 and 26 May 2021 to measure the crop’s biophysical variables (Table 3) with non-invasive methods. The leaf area index (LAI), fraction of absorbed photosynthetically active radiation (fAPAR), and fraction vegetation cover (fCover) were measured with an AccuPAR Ceptometer LP-80 (METER Group Inc., Pullman, WA, USA). Ten measurements evenly distributed over each plot (replication) were made and averaged, providing more representative vegetation canopy information.

Four representative plants were selected in the middle of each plot to measure leaf chlorophyll content (LCC) with a CCM-300 instrument (OPTISCIENCES). Chlorophyll content was measured in the middle of the flag leaf blade of these plants and averaged for each respective plot. The biophysical variables were only measured on the first and the second replication.

#### 2.2.3. Field Measurements of NDVI

Field measurements of NDVI (Table 4) were carried out during two field campaigns, in May and June 2021. The device used was a GreenSeeker Handheld Crop Sensor, Model HCS–100 [48]. During the calculation of NDVI, the red and NIR spectral bands with a central wavelength of 660 nm (25 nm full width at half maximum (FWHM) and 780 nm (25 nm FWHM) were used.

#### 2.2.4. UAV Data Collection

Two UAV platforms were used to obtain the multispectral data for the studied CVTs. These were a Sensefly eBee AG fixed-wing drone with a multispectral Parrot Sequoia camera (designated as PS in the following) and a DJI Phantom 4 Multispectral quadcopter (designated as P4M in the following). They were equipped with an integrated spectral sunlight sensor, which measured the sky down-welling irradiance and was used to retrieve the reflectance factors [49]. The cameras of both drones were provided with RGB and multispectral sensors. This study uses data from multispectral sensors. Both cameras featured green, red, red-edge, and NIR spectral bands (Table 5), but with different bandwidths. In addition, the P4M camera has a blue spectral band.

A total of four flight missions were carried out over the two CVTs (Table 6) during 2021. The flight missions were carried out within the period between 11:00 a.m. and 02:00 p.m. local time while observing the same parameters. The height was 100 m for PS and 50 m for P4M. The spectral images obtained by the PS featured a spatial resolution of 5 cm/pixel and those obtained by P4M were 2.5 cm/pixel. Both UAVs were equipped with a GPS, whereas the DJI Phantom 4 featured built-in real-time kinematic positioning (RTK). This provided a horizontal accuracy of ~5 m for the Sensefly eBee AG and 10–15 cm for DJI Phantom 4, respectively.

During the flight missions, the development stage was recorded (Table 6) using BBCH identification keys [50].

The small size of the plots (8 m × 1.25 m) requires achieving maximally high accuracy during the orthorectification of the UAV images. For this purpose, prior to the first flight mission, easily discernible objects were fixed, which were used as ground control points (GCPs). They were positioned at the four corners at CVT 1 and CVT 2, they were made of white material, and they constituted a square sized 20 m × 20 cm. They were fixed firmly on the terrain and were used during all flight missions. The geographical coordinates of the GCPs were measured with an accuracy of 1–3 cm by the GNSS equipment Leica GS08 and the RTK regime.

### 2.3. Image Processing and Data Extraction

The raw UAV images were processed using the photogrammetric software Pix4Dmapper (https://pix4d.com, accessed on 27 April 2023) to generate an orthophoto mosaic for every flight mission. The Pix4Dmapper software provides pre-defined camera models for both PS and P4M. It applies radiometric corrections to the images utilizing the spectral sunlight sensor’s data, which are recorded within the TIFF file header. The output represents the calculated reflectance and was provided as separate files for each spectral band. ENVI software was used to stack the bands into a multispectral image file; no additional adjustment was needed during the stacking because individual bands were aligned to each other by Pix4Dmapper. All mosaics were in the coordinate system UTM zone 35, datum WGS 1984. The multispectral mosaics were additionally geo-referenced in ArcGIS utilizing GCPs to achieve greater absolute geolocation precision and the correct overlapping of individual mosaics. A polygon layer with the boundaries of the plots was generated in ArcGIS by manually vectorizing them on an RGB image composite (from the first field campaign) used as a reference. The boundary of each plot was then subjected to a 10 cm inward buffering to avoid mixed pixels during data extraction. The buffered boundaries layer was used to extract the per plot-averaged reflectance for each spectral band, sensor, and date. The genotype name, replication number, and code of the plot were included in the attributive table for each polygon. In addition, spectral data were extracted from several bare soil areas, which were manually selected and vectorized in every mosaic.

### 2.4. Modeling and Statistical Analysis

The modeling process and statistical analysis are summarized in Figure 2.

The modeling process had two objectives: first, to retrieve GY from UAV multispectral data, and second, to retrieve the LAI, fAPAR, fCover, and LCC for the four replications for heritability analysis.

We trained parametric and nonparametric regression models using in situ and UAV multispectral data to retrieve GY at two development stages, milk ripening and dough ripening. The best GY model was applied to present a high-resolution map of the estimated GY. With the in situ measurements of the LAI, fAPAR, fCover, and LCC from only two replications, we trained nonparametric models for each biophysical variable retrieval. The best model for the LAI, fAPAR, fCover, and LCC was applied to the four replications plot for each genotype and used as remotely sensed phenotypic traits (RSPTs) for heritability analysis in crop selection.

The regression modeling was carried out with the ARTMO toolbox [51,52] (https://artmotoolbox.com/, accessed on 27 April 2023), version 3.29. Previous studies have shown that incorporating bare soil samples enhances the predictive power of models [53,54]; therefore, bare soil samples were included in the training data. The models were trained with training data, representing 2/3 of all the available data, and were optimized with tenfold cross-validation. The trained models were validated with validation data, representing 1/3 of all the available data, and were not used for training. The best-performing cross-validation models were selected according to the normalized root-mean-square-error (nRMSE). The selected metrics for the validation models are coefficients of determination (*R*^2^), root-mean-square-error (RMSE), nRMSE, relative RMSE (rRMSE), and Nash–Sutcliffe efficiency (NSE). The equations for these metrics are described in [55]. The goodness-of-fit metrics for the cross-validation were calculated by the ARTMO toolbox and the validation dataset in Python. Additionally, for GY retrieval, the Akaike information criterion (AIC) was calculated for each validation model as an additional tool for a comparison of the models [56].

Four widely used machine learning regression methods, partial least square regression (PLSR), random forest regression (RFR), kernel ridge regression (KRR), and Gaussian processes regression (GPR), and several generic types of vegetation indices (Table 7) and parametric functions, such as linear, exponential, logarithmic, power, and polynomial, were tested in this study.

PLSR [68] is a statistical technique used to identify the linear combinations of predictor variables that are highly correlated with a response variable. PLSR is especially useful in cases of noisy data and is computationally efficient; however, it is sensitive to outliers. RFR [69] is an ensemble machine learning method for regression problems. It combines multiple decision trees to produce a more accurate and stable prediction compared to a single decision tree. RFR can model both linear and non-linear relationships between the predictors and response variable; it is resistant to overfitting and robust to outliers. KRR [70] and GPR [71] are kernel-based methods. The kernel function used in our study was the radial basis function (RBF). The KRR model includes a regularization term (ridge regression) to prevent overfitting and a kernel function to transform the features into a higher-dimensional space. The kernel function maps the original features into a new space where the linear regression model can fit the data better and make more accurate predictions. The advantages of KRR include its ability to handle non-linear relationships between the predictors and response variables, its robustness to noisy data, and its ability to perform well on small datasets. GPR is a Bayesian machine learning technique. It models the target function as a Gaussian process, which is a collection of random variables of any finite number that have a joint Gaussian distribution. The advantages of GPR include its ability to make predictions with uncertainty estimates, its ability to handle non-linear relationships between the predictors and response variables, and its ability to adapt to changes in the underlying function as more data becomes available.

The phenotypic variance, including remotely sensed parameters, MGY, and ground-measured NDVI, was analyzed using a mixed linear model implemented in the R package “metan” that uses REML/BLUP (restricted maximum likelihood/best linear unbiased prediction) to estimate the variance components.

The function for analyzing single experiments (one-way experiments) using a mixed-effect model based on the following equation was used:(1)$$y_{ij}=\mu +\alpha_{i}+\tau_{j}+\varepsilon_{ij}$$
where $y_{ij}$ is the value observed for the *i*th genotype in the *j*th replicate (*i* = 1, 2, …, *g*; *j* = 1, 2, …, *r*); *g* and *r* being the number of genotypes and replicates, respectively; $\alpha_{i}$ is the random effect of the *i*th genotype; $\tau_{j}$ is the fixed effect of the *j*th replicate; and $\varepsilon_{ij}$ is the random error associated to $y_{ij}$.

Broad-sense heritability (H^2^) for each trait was calculated as the proportion of phenotypic variance explained by genetic variance [72].

Genotypic correlation (*rG*) among trait values was as follows:(2)$$rG=\frac{COVGxy}{\sqrt{\sigma^{2}Gx\cdot \sigma 2Gy}}$$
where *COVGxy* is the genotypic covariance of trait *x* and yield and *σ*^2^*Gx* and *σ*^2^*Gy* are the genotypic variances of trait *x* and yield, respectively.

Response to selection (*R*) of the RSPTs and grain yield and correlated response (*CR*) for grain yield by using RSPTs were calculated as:(3)$$R=h_{x}\cdot \sigma_{x}$$
where $h_{x}$ is the square root of heritability and *σ_x_* is the genotypic standard deviation:(4)$$CR=h_{x}\cdot rg\cdot \sigma_{y}$$
where $h_{x}$ is the square root of heritability of trait *x* (RSPTs), rg is the genetic correlation between the trait *x* (RSPTs) and *y* (MGY), and $\sigma_{y}$ is the genotypic standard deviation of trait *y* (MGY) [72]. The relative selection efficiency was calculated as the ratio of *CR* of grain yield for a specific RSPT and *R* of MGY [72].

## 3. Results

### 3.1. GY Retrieval

We trained models with the data using four replications per genotype from the four UAV missions independently to compare which development stage and camera were better suited for GY retrieval. The models were evaluated according to both cross-validation and validation metrics. The same (random) allocation of replicates among training and validation sets was used for all models.

The best-performing nonparametric model (Table 8 and Figure 3) was GPR with data from the May 2021 campaign and DJI Phantom 4/P4M (Mission2), with an *R*^2^ validation of 0.61, as shown in Figure 4. This model had, with the validation dataset, the lowest RMSE (1034.85 kg/ha) and good rRMSE (16.01%) and NSE (0.60) [55]. It was followed by FRF with data from the May 2021 campaign and eBee Ag/PS (Mission1), with an *R*^2^ validation of 0.45. The models with data from the June 2021 campaign performed poorly with the validation data, with an *R*^2^ validation of 0.33 for eBee Ag/PS (Mission3) and 0.36 for DJI Phantom 4/P4M (Mission4).

The best-performing parametrical model (Table 9) was with two spectral bands index, EVI2-like with Ra = 730 and Rb = 840, linear function and data from the May 2021 campaign and DJI Phantom 4/P4M (Mission2). The model achieved an *R*^2^ validation of 0.45, which was closely followed by the model with May 2021 campaign with eBee Ag/PS (Mission1), with an *R*^2^ validation of 0.44. However, both models had the same value as AIC, which indicates the ability to fit the data while avoiding overfitting. Therefore, we can consider both models equal in performance for the validation dataset.

The nonparametric models performed better than the parametric models independently of the retrieval method and camera used, the milk ripening development stage, or May campaign, which were better suited for GY retrieval (Table 7 and Table 9) than the dough ripening or June campaign. This was clear from the AIC values of the parametric models in Mission1 and Mission2, which were higher than the AIC values of the nonparametric models in Mission3 and Mission4.

### 3.2. Biophysical Variable Retrieval

We trained nonparametric models using data from two replications per genotype and data from a UAV mission that produced the best GY estimation results, specifically milk ripening during Mission2 (DJI Phantom 4/P4M). The models were evaluated using both cross-validation and validation metrics. The best results from the LAI, fAPAR, fCover, and LCC retrieval are presented in Table 10 and Figure 5.

The evaluation of models is complicated task because the variables being analyzed may have differences in magnitude, range, mean, and variance [55].

The LAI retrieval model had the highest performance, with a validation *R*^2^ of 0.71. The validation RMSE of 0.56 m^2^ m^−2^ was also close to the recommended excellent result for the LAI [55], and the rRMSE and NSE values were good. The fAPAR and fCover retrieval models had validations *R*^2^ of 0.52 and 0.48, respectively. The LLC retrieval model, on the other hand, only obtained a validation *R*^2^ of 0.44.

The best model for fAPAR and fCover was GPR, which provides uncertainty estimates. Therefore, an uncertainty estimation was generated by each GPR model. The uncertainty for the validation dataset for fAPAR was under 5% and for fCover, it was under 6%.

The best-performing model per biophysical variable, as shown in Table 10, was then used to retrieve four replications per genotype for the LAI, fAPAR, fCover, and LCC (Figure 6). The modeled biophysical variables are used as RSPTs for the analysis of broad-sense heritability.

### 3.3. Broad-Sense Heritability

A significant effect on the genotype for all evaluated traits according to likelihood ratio validation (Table 11), indicating genetic variability between barley genotypes, was found.

High heritability estimates were found for MGY and RSPTs in CVT2 and pooled data (Table 12). CVT1, MGY, NDVI_1, and NDVI_2 had moderate heritability values, while RSPTs had high heritability. MGY heritability was lower, with a few exceptions compared to RSPTs, indicating that MGY was more environmentally influenced than RSPTs. Considerably higher heritability estimates were found for the LAI, fAPAR, fCover, and LCC than ground-measured NDVI.

#### 3.3.1. Genetic Correlation

The studied RSPTs showed statistically significant genetic correlations with MGY in CVT2 on the basis of all tested genotypes (pooled data) (Table 13). Insignificant correlations of MGY with NDVI_2 and LCC were found in CTV1. The estimated correlation coefficients for CVT2 were higher compared to those of CVT1 and varied from *r* = 0.459 for the LAI to *r* = 0.686 for NDVI_1.

#### 3.3.2. Correlated Response of Grain Yield and Efficiency of Indirect Selection

MGY showed the highest response to direct selection (Table 14). Overall, in pooled data, the LAI had the highest correlated response, resulting in the highest indirect selection efficiency for MGY. NDVI_2 gave the lowest correlated response for MGY.

## 4. Discussion

### 4.1. Grain Yield and Biophysical Variable Modeling

This study investigated the potential of two consumer-grad multispectral UAV sensors, PS and P4M, for predicting GY and retrieving four biophysical variables (LAI, fAPAR, fCover, and LCC) in winter barley trails. Both sensors have been previously tested for GY assessment in different crops, providing reasonable results [73,74,75,76].

Previous studies observed large fluctuations in the prediction power for remote sensing-based GY models obtained at different time points of the vegetation period. For example, Elsayed et al. [29] utilized PLSR to predict GY achieving *R*^2^ between 0.11 and 0.52. The predictors were a set of spectral indices calculated from hyperspectral sensor data. The different models were calibrated with data collected at growth stages BBCH 55 and 60 and under mild and severe drought stress conditions. In two related studies (using the same hyperspectral sensor and PLSR), Rischbeck et al. [30] achieved an *R*^2^ of up to 0.65 for a model calibrated at growth stage BBCH 59, while Barmeier et al. [31] achieved an *R*^2^ varying between 0.71 and 0.95 for a model at anthesis. A tendency for increasing the accuracy of prediction with the advance of crop development is usually reported in the literature [9,27]. Yield modeling in this study was performed at two relatively late time points, at the milk ripening and dough ripening stages, respectively. We observed that GY was predicted with better accuracy in the milk ripening stage. The dough ripening, on the other hand, appeared to be too late for accurate prediction, probably because spectral differences between genotypes became less pronounced with the senescence. The retrieval of agronomic parameters through remote sensing usually requires high variability in the modeling calibration dataset for optimal results [76]. Moreover, we used data from a single field campaign for the models. Combining data from several time points, or different treatments, during the vegetation period may improve modeling results [75,76].

The nonparametric models performed better than the parametric ones in predicting GY. This result was in agreement with other studies, which have shown the superiority of machine learning algorithms over VI-based regression models for predicting crop traits [77,78].

During the field campaigns, we collected image data from both UAV sensors on the same day and under similar illumination conditions (clear sky) to facilitate their comparison. Based on the results of the present study, we can conclude that the P4M sensor was better suited for assessing GY with nonparametric models (Table 8). However, when using parametric models (Table 9), no conclusion can be drawn about the relative advantages of the sensors in assessing GY. Nevertheless, the most accurate model of GY in this study was based on P4M data. It should be noted that the nonparametric models utilized the full spectral information from a sensor (all bands), whereas the parametric ones usually relied on three-band and two-band VIs, or even single-band reflectance. Therefore, results from the nonparametric modeling may provide a more integrated assessment of the full potential of the sensor. The main differences between the sensors with respect to spectral bands consist of the lack of a blue band in PS and the different width and central wavelength of the red-edge and near-infrared bands (Table 5).

The retrieval accuracy varied considerably between the biophysical variables. The prediction was the most successful for the LAI, while LCC was the most challenging to estimate using UAV data. The trial in this study was conducted in rainfed conditions and there were no treatments of varying agronomic management (fertilization, irrigation, etc.). The lack of variation in growing conditions was expected to result in lower variation in GY and biophysical variables among genotypes and, therefore, lower correlations, as discussed earlier. In particular, the lower coefficients of determination for fAPAR, fCover, and LCC models, compared to the LAI, could be partially attributed to the low variability of these variables (Table 3). The LAI was modeled with good accuracy in this study (*R*^2^ = 0.71; nRMSE = 13%). Moreover, the reported accuracy was in the validation dataset and was not included in the training, whereas in other studies [79], the reported accuracy was only in the cross-validation dataset. Furthermore, the accuracy could be improved by adding canopy surface height [80,81] and texture variables [80] as predictors.

### 4.2. Broad-Sense Heritability and the Application of Remotely Sensed Traits for Breeders

The genetic correlation describes the genetic relationship between two traits and provides a measure of the rate at which traits respond to indirect selection. The presence of a sufficiently high level of genetic diversity is a necessary condition for an accurate estimation of this association. Accordingly, the included CVTs in our experiment represented the winter barley varieties with different origins and the most advanced breeding lines from the breeding program at the Institute of Agriculture, Karnobat. Furthermore, a significant genotypic effect was found for all studied RSPTs and measured grain yield. The genetic correlations between grain yield and most of the RSPTs were consistently significant across two sets of tested genotypes, demonstrating the reliability of these estimates.

The moderate to strong genetic correlation of RSPTs to grain yield in the present study indicated the potential use of RSPTs as an indirect selection approach to identify high-yield genotypes of winter barley. The improvement of grain yield is a primary goal for most barley breeding programs, but the complex nature of this trait complicates breeding and the selection of lines with high yield potential. The effectiveness of indirect selection depends on using secondary traits that have a genetic correlation with grain yield and higher heritability than yield. Spectral reflectance indices have been utilized as such traits for the indirect selection of wheat lines with high yield potential [8,43,44,82,83,84].

The studied RSPTs had generally higher heritability than grain yield. Despite that, all RSPTs had lower efficiency of indirect over the direct selection for grain yield. Gizaw et al., 2016 [44] point out that spectral reflectance can be used in plant breeding, not just as a standalone indirect selection criterion, but also as a component of an integrated selection approach. According to these authors, integrated selection of grain yield and spectral reflectance indices increased the repeatability of genotypic performance across multiple trials.

Moreover, in addition to genetic gain, factors such as phenotyping speed and cost are important for selection efficiency [85]. The remotely sensed phenotyping allows screening of a larger set of germplasm than yield-based selection. Manneveux and Ribaut 2006 [85] proposed using lower selection intensity for selection based on spectral reflectance indices so that most superior genotypes could be advanced through yield-based selection, which increased the indirect selection efficiency substantially.

### 4.3. Limitations, Challenges, and Future Opportunities

This research demonstrated a possible use of remote sensing techniques and data in plant breeding experiments through the successful use of both parametric and nonparametric regression models to retrieve yield and remotely sensed phenotypic traits in winter barley. However, there are some limitations and challenges that are outlined below, along with suggested solutions.
(1)Remotely sensed phenotypic traits are part of the next-generation physiological breeding research [86]. However, the appropriate range of goodness-of-fit metrics for biophysical variables models to be utilized as RSPTs is yet to be established. Additionally, conducting an uncertainty analysis on the RSPTs used as input data in plant breeding analysis is a critical aspect that should be routinely implemented. Propagating uncertainty from the acquisition of in situ and remotely sensed data to the modeling stage will enable plant breeders to gain a comprehensive understanding of the precision and accuracy of the proposed RSPTs.(2)In our current study, we analyzed UAV imagery with four and five spectral bands. Our findings indicate that the five-band imagery resulted in better models, suggesting that further testing of multispectral and hyperspectral UAV imagery is needed to obtain the optimal number and type of bands that may yield improved retrieval models [87].(3)The current models have been trained and evaluated using data from one growing year, specifically from the breeding lines of the Institute of Agriculture, Karnobat breeding program. While they have shown promising results, the robustness of these models throughout multiple growing seasons still needs to be evaluated.(4)The selection environment is crucial to assessing the genetic gain and efficiency of alternative selection approaches [85]. Therefore, for the complete assessment of the efficiency of RSPTs for the indirect selection of high-yielding barley lines, additional studies under different environments (growing years and locations) are needed.

## 5. Conclusions

The primary goal of a breeding program is to develop cultivars with high yields and superior quality that can be released to farmers. To achieve this objective, it is essential (1) to estimate GY as soon as possible during the growing season and (2) to identify remotely sensed traits that could have a role in the broad-sense heritability analysis. Data from DJI Phantom 4 Multispectral at the milk ripening stage predicted barley GY most accurately. Biophysical variables, the LAI, fAPAR, fCover, and LCC have been retrieved with data from DJI Phantom 4 Multispectral at the milk ripening stage and are used as RSPTs for broad band heritability. They had generally higher heritability than grain yield but lower efficiency over the direct selection for grain yield. However, the acceptable range of goodness-of-fit metrics for biophysical variables models to be used as RSPTs is still to be determined.

We are of the opinion that these initial findings have the potential to accelerate crop improvement programs. However, further robust interdisciplinary research [88] is still required to strengthen these results.

## Figures and Tables

**Figure 1 sensors-23-05008-f001:**
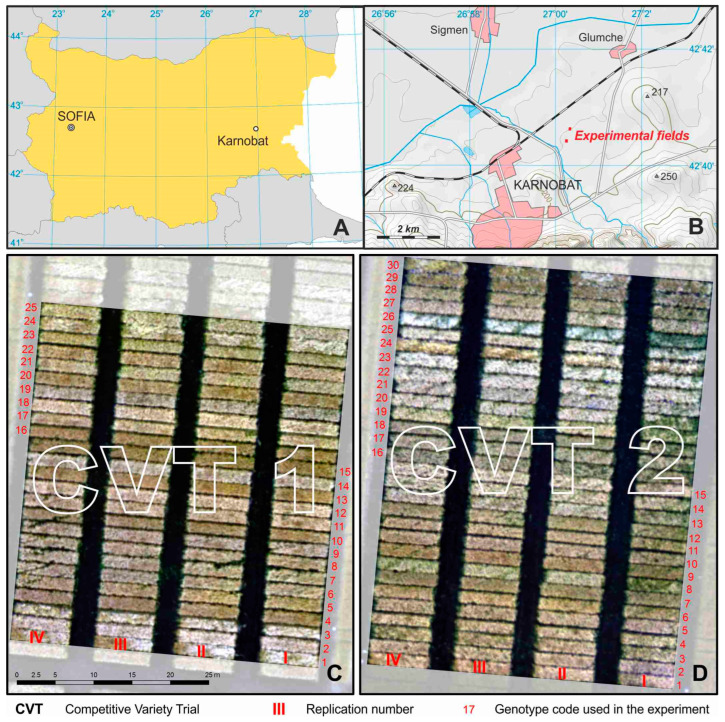
Study area. (**A**) Location of the Institute of Agriculture, Karnobat, Bulgaria. (**B**) Location of the breeding experimental fields. (**C**) Orthophoto map of the competitive variety trial *CVT 1,* in which 25 winter barley genotypes were sown with 4 replicates. (**D**) Orthophoto map of the competitive variety trial *CVT 2,* in which 30 winter barley genotypes were sown with 4 replicates.

**Figure 2 sensors-23-05008-f002:**
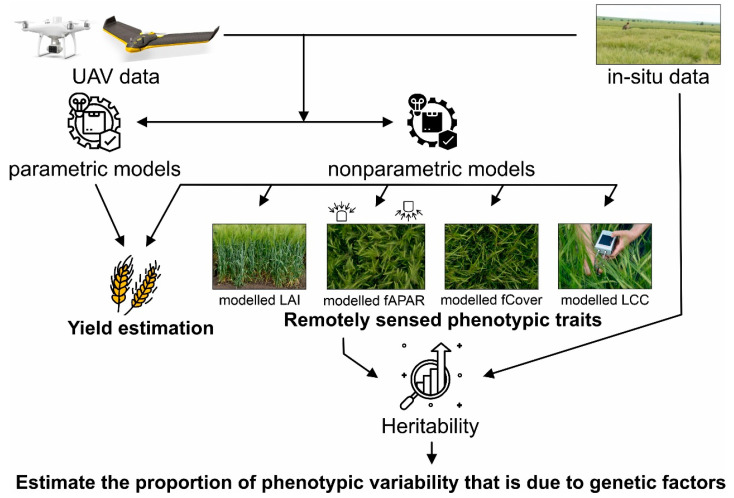
Workflow diagram of phenotypic traits estimation and preliminary yield assessment in the barley breeding experiment. The results of the study are in bold.

**Figure 3 sensors-23-05008-f003:**
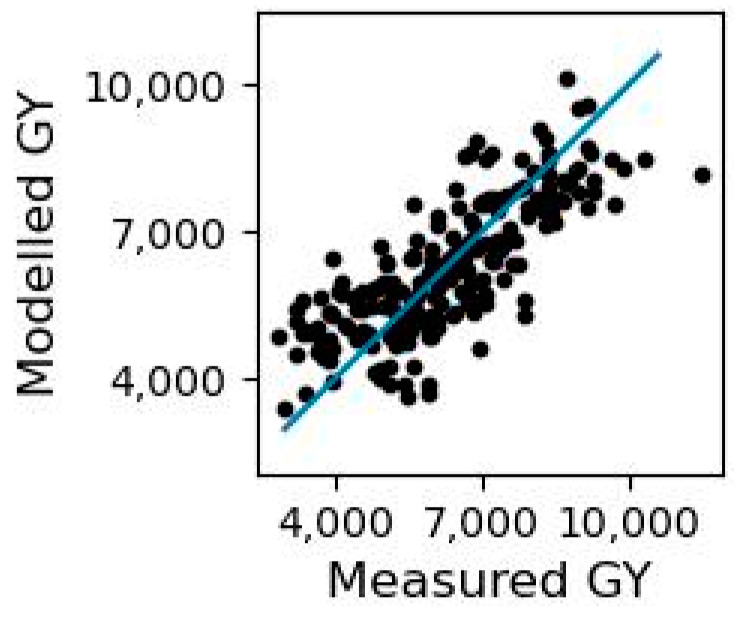
Relationship between the measured and modeled GY (kg/ha) with nonparametric GPR model and data from Mission2 for all available vegetation samples (*n* = 220) without the soil samples. The blue line is the 1:1 line.

**Figure 4 sensors-23-05008-f004:**
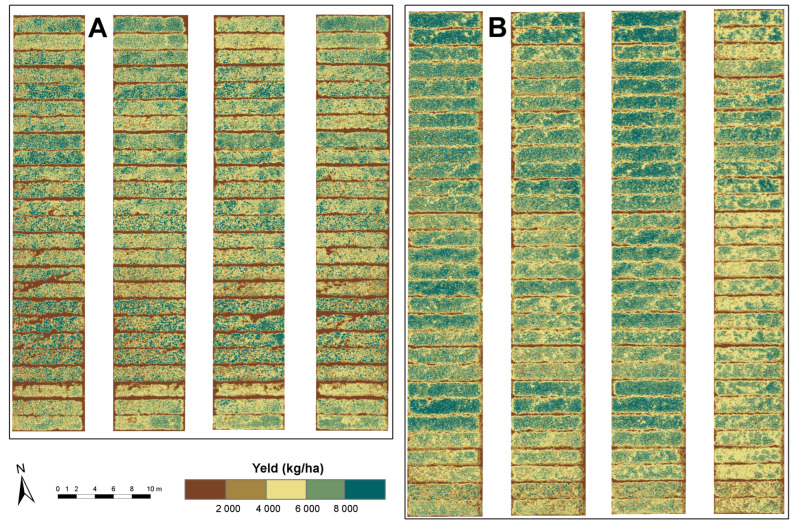
Map of estimated GY for CVT-1 (**A**) and CVT-2 (**B**) for the milk ripening (May 2021) development stage. The GY was estimated with the GPR model and data from Mission2.

**Figure 5 sensors-23-05008-f005:**
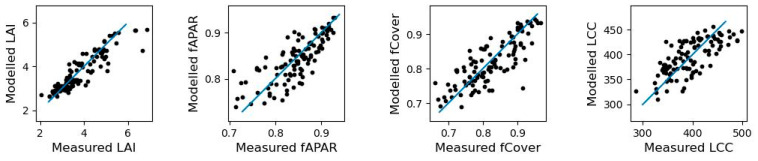
Relationship between the measured and modeled biophysical variables for all available vegetation samples (*n* = 108) without the soil samples. The blue line is the 1:1 line.

**Figure 6 sensors-23-05008-f006:**
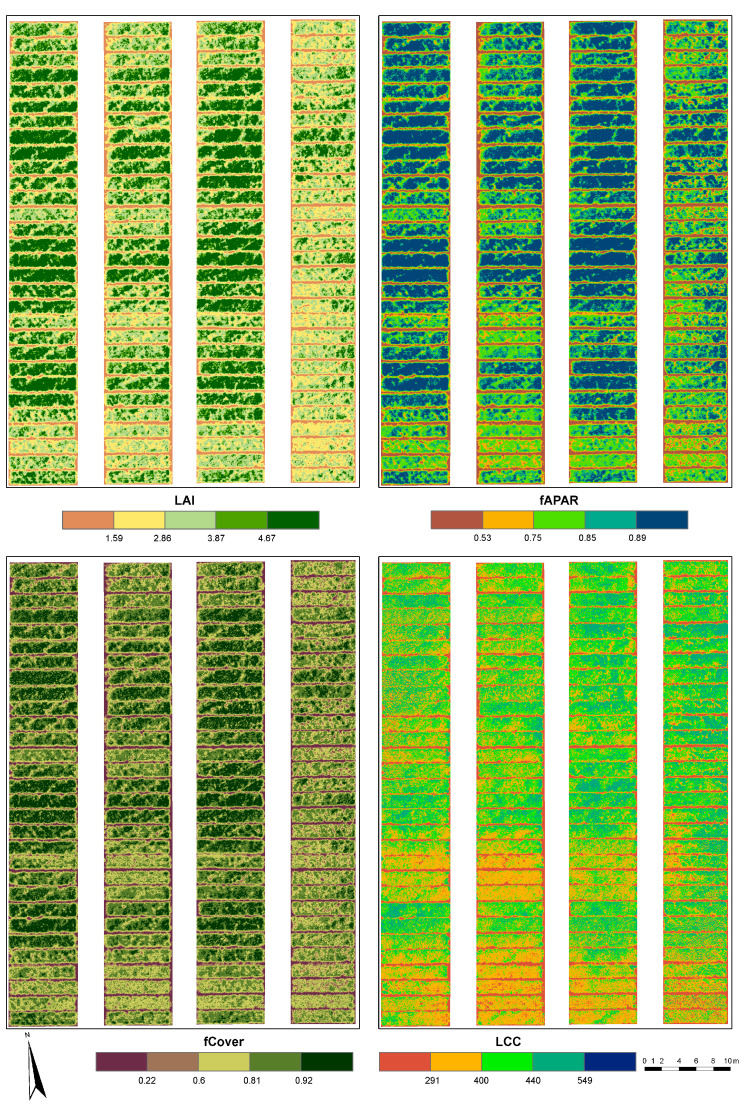
UAV-based (Mission2) maps of the estimated LAI, fAPAR, fCover, and LCC for CVT-2 for the milk ripening (May 2021) development stage. The LAI was estimated with the RFR model. The fAPAR and fCover were estimated with the GPR model. LCC was estimated with the KRR model.

**Table 1 sensors-23-05008-t001:** Average air temperature, monthly sums of precipitation, and long-term average data in Karnobat, Southeastern Bulgaria during barley vegetation. T—average air temperature; P—the sum of precipitation; LT—long-term average air temperature and the sum of precipitation (1931–2022).

Months	2020/2021		LT	
	T, °C	P, mm	T, °C	P, mm
X	15.8	70.7	12.5	44.3
XI	7.4	25.5	7.1	53.7
XII	6.6	94.5	2.6	51.2
I	3.7	142.8	0.6	36.5
II	4.9	22.1	2.2	35.8
III	4.9	47.4	5.3	34.1
IV	9.5	86	10.5	45.3
V	16.6	15.6	15.6	58.5
VI	19.3	117.4	19.6	65.2
T, °C	9.9		8.4	
P, mm		622.0		424.6

**Table 2 sensors-23-05008-t002:** Descriptive statistics for the measured grain yield (kg/ha) of winter barley per the competitive variety trial (CVT) and combined.

CVT	Number of Measurements	Min. Value	Max. Value	Mean	Std. Dev.	CV%
CVT 1	100	2860	11,460	5228	1432	27.39
CVT 2	120	4240	10,300	7155	1349	18.85
Total	220	2860	11,460	6279	1686	26.85

**Table 3 sensors-23-05008-t003:** Descriptive statistics for the ground-measured biophysical variables of winter barley during the field campaigns conducted on 25 and 26 May 2021. CVT = competitive variety trial. fAPAR and fCover are expressed as unitless fractions. The values range from 0 to 1, where 0 represents no vegetation cover or an absorption of radiation and 1 represents complete vegetation cover or an absorption of all incident radiation.

Biophysical Variables	CVT	Number of Measurements	Min. Value	Max. Value	Mean	Std. Dev.	CV%
LAI [m^2^ m^−2^]	CVT 1	50	2.55	6.87	3.65	0.80	21.92
fAPAR		50	0.71	0.92	0.82	0.05	6.10
fCover		50	0.67	0.96	0.79	0.07	8.86
LCC [mg m^−2^]		50	285.75	435.00	363.70	29.35	8.07
LAI [m^2^ m^−2^]	CVT 2	58	2.05	6.65	3.92	1.10	28.06
fAPAR		58	0.71	0.93	0.86	0.04	4.65
fCover		58	0.66	0.97	0.85	0.07	8.24
LCC [mg m^−2^]		58	345.75	498.00	420.00	38.15	9.08
LAI [m^2^ m^−2^]	Total	108	2.05	6.87	3.79	0.98	25.86
fAPAR		108	0.71	0.93	0.85	0.05	5.88
fCover		108	0.66	0.97	0.82	0.08	9.76
LCC [mg m^−2^]		108	285.75	498.00	393.93	44.33	11.25

**Table 4 sensors-23-05008-t004:** Descriptive statistics for the ground-measured NDVI of winter barley. CVT = competitive variety trial.

Date/Development Stage	Identifier	CVT	Number of Measurements	Min. Value	Max. Value	Mean	Std. Dev.	CV%
25 May 2021/Milk ripening	NDVI_1	CVT 1	100	0.53	0.77	0.68	0.05	7.35
		CVT 2	120	0.58	0.81	0.71	0.04	5.63
		Total	220	0.53	0.81	0.7	0.05	7.14
14 June 2021/Dough ripening	NDVI_2	CVT 1	100	0.1	0.25	0.17	0.03	17.65
		CVT 2	120	0.1	0.59	0.31	0.1	32.26
		Total	220	0.11	0.59	0.25	0.11	44

**Table 5 sensors-23-05008-t005:** Bands’ spectral ranges (nm) of the used multispectral UAV sensors: PS (eBee AG/Sequoia) and P4M (DJI Phantom 4 Multispectral).

Band	PS	P4M
Blue (B)	-	434–466
Green (G)	530–570	544–576
Red (R)	640–680	634–666
Red-edge (RE)	730–740	714–746
Near-infrared (NIR)	770–810	814–866

**Table 6 sensors-23-05008-t006:** UAV flight missions’ dates, corresponding development stages of the studied genotypes of winter barley, and weather conditions during the flight.

Flight Mission/ID	UAV/Camera	Flight Date	Development Stage	Weather Condition
Mission1/M1	eBee Ag/Sequoia	26 May 2021	BBCH 73, BBCH 75, BBCH 77, BBCH 81	Clear sky
Mission2/M2	DJI Phantom 4/Multispectral			
Mission3/M3	eBee Ag/Sequoia	15 June 2021	BBCH 75, BBCH 77, BBCH 81, BBCH 83, BBCH 85, BBCH 87	Clear sky
Mission4/M4	DJI Phantom 4/Multispectral			

**Table 7 sensors-23-05008-t007:** List of the generic types of vegetation indices selected for testing in the study.

Name	Formula ^1^	Related to	Reference
R	Ra	Chl	[57]
SR	Ra/Rb	Chl, Chl, LAI, GY	[57,58,59,60]
DVI	Ra − Rb	Chl	[57]
ND	(Ra − Rb)/(Ra + Rb)	GY, fAPAR, fCover	[58,59,60]
mSR	(Ra − Rc)/(Rb − Rc)	Chl	[57,58]
mSR2	(Ra/Rb) − 1	GY, Chl	[59,61]
mND	(Ra − Rb)/(Ra + Rb – 2 × Rc)	Chl, LAI	[57]
3SBI-Verrelst	(Ra − Rc)/(Rb + Rc)	LAI	[52]
3SBI-Tian	(Ra − Rb − Rc)/(Ra + Rb + Rc)	LAI	[62]
3SBI-Wang	(Ra − Rb + 2 × Rc)/(Ra + Rb – 2 × Rc)	Chl, LAI	[63]
3SBI-Dash	(Ra − Rb)/(Rb − Rc)	Chl	[64]
4SBI	((Ra − Rb)/(Ra + Rb))/((Rc − Rd)/(Rc + Rd))	Above ground dry biomass	[58]
EVI-like	2.5 × ((Rb − Rc)/(Rb + 6 × Rc + 7.5 × Ra + 1))	Chl, LAI, fCover, GY	[65,66]
EVI2-like	2.5 ×((Ra − Rb)/(Ra + 2 × Rb + 1))	GY	[67]

^1^ Ra, Rb, Rc, and Rd represent reflectance at different wavelengths.

**Table 8 sensors-23-05008-t008:** Results of GY retrieval with PS (eBee Ag/Sequoia) and P4M (DJI Phantom 4 Multispectral) data with best nonparametric models. *n*_training = 207, *n*_validation = 73.

Date/Sensor	Flight Mission	Model	Cross-Validation						Validation					
			*R* ^2^	RMSE (kg/ha)	nRMSE (%)	rRMSE (%)	NSE		*R* ^2^	RMSE (kg/ha)	nRMSE (%)	rRMSE (%)	NSE	AIC
26 May 2021/PS	Mission1	RFR	0.88	1086	9.48	24.73	0.88		0.45	1209.66	17.16	18.72	0.44	1044
26 May 2021/P4M	Mission2	GPR	0.90	995	8.68	22.64	0.90		0.61	1034.85	14.68	16.01	0.60	1024
15 June 2021/PS	Mission3	GPR	0.85	1232	10.75	28.03	0.85		0.33	1348.02	19.12	20.86	0.31	1060
15 June 2021/P4M	Mission4	PLSR	0.86	1188	10.37	27.04	0.86		0.36	1305.67	18.52	20.20	0.35	1057

**Table 9 sensors-23-05008-t009:** Results of GY retrieval with PS (eBee Ag/Sequoia) and P4M (DJI Phantom 4 Multispectral) data with best parametric models. *n*_training = 207, *n*_validation = 73.

Date/Sensor	Flight Mission	VI (Bands)/Function	Cross-Validation						Validation					
			*R* ^2^	RMSE (kg/ha)	nRMSE (%)	rRMSE (%)	NSE		*R* ^2^	RMSE (kg/ha)	nRMSE (%)	rRMSE (%)	NSE	AIC
26 May 2021/PS	Mission1	DVI (550,790)/Polynomial	0.88	1102	9.62	25.09	0.88		0.44	1224.79	17.37	18.95	0.43	1046
26 May 2021/P4M	Mission2	EVI2-like (730,840)/Linear	0.86	1171	10.22	26.65	0.86		0.45	1207.79	17.13	18.95	0.44	1046
15 June 2021/PS	Mission3	R (790)/Linear	0.84	1273	11.11	28.98	0.84		0.31	1372.61	19.47	21.24	0.28	1063
15 June 2021/P4M	Mission4	EVI-like (730,840,450)/Linear	0.85	1239	10.81	28.20	0.85		0.26	1409.59	19.99	21.81	0.24	1069

**Table 10 sensors-23-05008-t010:** Results of best models for biophysical variable retrieval with P4M (DJI Phantom 4 Multispectral) data of 26 May 2021. *n*_training = 102, *n*_validation = 36.

Date/Sensor	Flight Mission	Biophysical Variable	Model	Cross-Validation						Validation				
				*R* ^2^	RMSE	nRMSE (%)	rRMSE (%)	NSE		*R* ^2^	RMSE	nRMSE (%)	rRMSE (%)	NSE
26 May 2021/P4M	Mission2	LAI	RFR	0.95	0.42	6.09	15.53	0.95		0.71	0.56	13.29	14.96	0.69
		fAPAR	GPR	0.99	0.03	3.32	5.12	0.99		0.52	0.04	19.61	4.27	0.35
		fCover	GPR	0.99	0.04	4.14	6.79	0.99		0.48	0.04	20.4	6.97	0.41
		LCC	KRR	0.98	27.74	5.57	9.82	0.98		0.44	33.66	16.50	8.82	0.38

**Table 11 sensors-23-05008-t011:** The likelihood ratio test for grain yield and different remotely sensed phenotypic traits in two CVTs of winter barley genotypes.

CVT	MGY	NDVI_1	NDVI_2	LAI	fAPAR	fCover	LCC
CVT1	12.03 *	34.02 *	15.51 *	97.16 *	137.43 *	117.95 *	105.85 *
CVT2	70.01 *	51.65 *	64.45 *	72.48 *	139.48 *	83.75 *	121.16 *

* All variables with a significant (*p* < 0.05) genotype effect.

**Table 12 sensors-23-05008-t012:** Broad-sense heritability (H^2^) of grain yield and different remotely sensed phenotypic traits in two CVTs of winter barley genotypes.

CVT	MGY	NDVI_1	NDVI_2	LAI	fAPAR	fCover	LCC
CVT1	0.33	0.55	0.33	0.83	0.92	0.88	0.85
CVT2	0.69	0.63	0.67	0.70	0.83	0.75	0.84
Pooled	0.63	0.61	0.66	0.80	0.86	0.80	0.87

**Table 13 sensors-23-05008-t013:** Genotypic correlation between grain yield and different remotely sensed phenotypic traits in two CVTs of winter barley genotypes.

CVT	NDVI_1	NDVI_2	LAI	fAPAR	fCover	LCC
CVT1	0.437 *	−0.223 ns	0.404 *	0.432 *	0.468 *	0.127 ns
CVT2	0.686 **	0.560 **	0.459 *	0.582 **	0.491 **	0.663 **
Pooled	0.689 **	0.761 **	0.408 **	0.620 **	0.468 **	0.556 **

* Significant at the 0.05 probability level. ** Significant at the 0.01 probability level. ns: non-significant.

**Table 14 sensors-23-05008-t014:** Selection response (*R*) for MGY and remotely sensed phenotypic traits, correlated response (*CR*) for grain yield using remotely sensed phenotypic traits, and relative selection efficiency (*CR/R*) of the remotely sensed phenotypic traits for grain yield in the pooled data from two CVTs of winter barley genotypes.

Parameters	MGY	NDVI_1	NDVI_2	LAI	fAPAR	fCover	LCC
R	1020.15	0.03	0.48	0.09	0.04	0.05	27.29
CR		548.30	338.73	693.82	587.78	427.03	527.74
CR/R		0.54	0.33	0.68	0.58	0.42	0.52

## Data Availability

Not applicable.

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
