# Peer review of "Remotely Sensed Phenotypic Traits for Heritability Estimates and Grain Yield Prediction of Barley Using Multispectral Imaging from UAVs"

_sensors, 2023, doi:10.3390/s23115008_

Round 1

Reviewer 1 Report

The english language should be improved.

Author Response

Dear reviewer,

thank you for your time and comments/suggestions.

Please find in the attached file our response to your comments/suggestions.

Best

Reviewer 2 Report

This manuscript (sensors-2398434) explored the use of parametric and nonparametric regression models with multispectral data from UAVs to predict grain yield in barley breeding experiments. Nonparametric models outperformed parametric models, with the highest accuracy achieved during the milk ripening stage using DJI Phantom 4 multispectral images, indicating potential for indirect selection of high-yielding genotypes.

- provide full spelling of all acronyms appearing in the abstract.

“combination?

several replications?

1 200 and 2 500, not correct!

Units in fAPAR and fCover?

Meteorological data should be incorporated into the manuscript to provide a more comprehensive understanding of the environmental factors influencing the study's results.

The last sentence of the conclusion can be omitted as it does not add significant value to the overall understanding of the study.

The R2 values are low, and the models presented for grain yield prediction and phenotyping using multispectral images are not effective despite the authors' efforts. Although the potential for using multispectral imagery exists, the study did not successfully identify high-yielding genotypes through this method.

In Figure 1, there is no need to include CT1 or CVT2 in the image, as they may not contribute significantly to the illustration or understanding of the presented data.

I would like to understand how multispectral data can predict biophysical parameters, such as LAI and biomass? How does spectral behaviour correlate with these predictions? Could it be a spurious correlation?

Figures 3 and 5. What is the sample size between measured and modelled (predicted) values?

The paragraphs in the discussion 4.1 section are quite lengthy; consider shortening them and removing verbosity to improve clarity and readability for the reader.

Although the manuscript is interesting, my primary concern is the generated models with low regression coefficients. The models also cannot accurately estimate biophysical attributes. There is no selection of more specific wavelengths. Moreover, while NDVI is a widely used vegetation index, it is not capable of providing properties for estimation as presented in the manuscript.

Best regards

Major corrections in grammar and spelling are needed. Additionally, many paragraphs need to be shortened for better clarity.

Author Response

(The authors gave the same response as above.)

Reviewer 3 Report

This is a wonderful research using Unmanned Aerial Vehicle Multi spectral Imaging Technology. I recommend this manuscript for publication, the only small suggestion is to enlarge the photos in Figure 2.

Author Response

(The authors gave the same response as above.)

Round 2

Reviewer 1 Report

The author has revised it in accordance with my comments.

Reviewer 2 Report

I thank the authors for considering my points and questions. Many issues have been clarified and the manuscript has been improved.

grammar and spelling check.